# Dynamic Image for 3D MRI image Alzheimer's Disease classification

**Abstract.** We propose to apply a 2D CNN architecture to 3D MRI image Alzheimer's disease classification. Training a 3D convolutional neural network (CNN) is time-consuming and computationally expensive. We make use of dynamic image technology to transform the 3D MRI image volume into a 2D image to use as input to a 2D CNN. We show our proposed CNN model achieves 9.5% better Alzheimer's disease classification accuracy than the baseline 3D models. We also show that our method allows for efficient training, requiring only 20% of the training time compared to 3D CNN models. The code is available online: https://github.com/xxx/xxx.

**Keywords:** Dynamic image, 2D CNN, MRI image, Alzheimer's Disease

## 1 Introduction

Alzheimer's disease (AD) is the sixth leading cause of death in the U.S. [1]. It heavily affects the patients' families and U.S. health care system due to medical payments, social welfare cost, and salary loss. Since AD is irreversible, early stage diagnosis is crucial for helping slow down disease progression. Currently, researchers are using advanced neuroimaging techniques, such as magnetic resonance imaging (MRI), to identify AD. MRI technology produces a 3D image, which has millions of voxels. Figure 1 shows example slices of Cognitive Unimpaired (CU) and Alzheimer's disease (AD) MRI images.

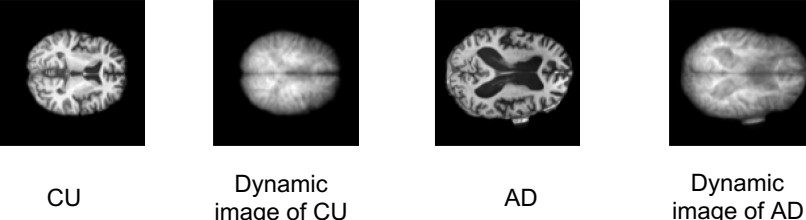

| CU | Dynamic image of CU | AD | Dynamic image of AD |

**Fig. 1.** The MRI sample slices of the CU and AD participants and the corresponding dynamic images. The first row images label are CU and the second row images label are AD.

With the promising performance of deep learning in natural image classification, convolutional neural networks (CNNs) show tremendous potential in medical image diagnosis. Due to the volumetric nature of MRI images, the natural deep learning model is a 3D convolutional neural network (3D CNN) [10]. Compared to 2D CNN models, 3D CNN models are more computationally expensive and time consuming to train due to the high dimensionality of the input. Another issue is that most current medical datasets are relatively small. The limited data makes it difficult to train a deep network that generalizes to high accuracy on unseen data. To overcome the problem of limited medical image training data, transfer learning is an attractive approach for feature extraction. However, pre-trained CNN models are mainly trained on 2D image datasets. There are few suitable pre-trained 3D CNN models. In our paper, We propose to apply dynamic images [3] to convert a 3D MRI volume into a 2D image over the height dimension. Thus, we can use a 2D CNN architecture for 3D MRI image classification. The main contributions of our work are following:

- We propose to apply a CNN model that transforms the 3D MRI volume image into 2D dynamic image as the input of 2D CNN. Incorporating with an attention mechanism, the proposed model significantly boosts the accuracy of the Alzheimer's Disease MRI diagnosis.
- We conduct the preliminary experiments on an efficient network [9] as the feature extractor. Under appropriate network design, our method has the potential for mobile applications of 3D medical image diagnosis.
- We analyze the effect of skull MRI images on the dynamic image method, showing that the applied dynamic image method is sensitive to the noise introduced by the skull. Skull striping is necessary before using the dynamic image technology.

## 2   Related Work

Learning-based Alzheimer's disease (AD) research can be mainly divided into two branches based on the type of input: (1) manually selected region of interest (ROI) input and (2) whole image input. With ROI models [6] [14], manual region selection is needed to extract the interest region of the original brain image as the input to the CNN model, which is a time consuming task. It is more straightforward and desirable to use the whole image as input. Korolev et al. [11] propose two 3D CNN architectures based on VGGNet and ResNet, which is the first study to prove the manual feature extraction step for Brain MRI image classification is unnecessary. Their 3D models are called 3D-VGG and 3D-ResNet, and are widely used for 3D medical image classification study. Cheng et al. [4] proposes to use multiple 3D CNN models trained on MRI images for AD classification in an ensemble learning strategy. They separate the original MRI 3D images into many patches (n=27), then forward each patch to an independent 3D CNN for feature extraction. Afterward, the extracted features are concatenated for classification. The performance is satisfactory, but the computation cost and training time overhead are very expensive. Yang et al. [18]

uses the 3D-CNN models of Korolev et al. [11] as a backbone for studying the explainability of AD classification in MRI images by extending class activation mapping (CAM)[20] and gradient-based CAM[16] on 3D images. In our work, we use the whole brain MRI image as input and use 3D-VGG and 3D-ResNet as our baseline models.

Dynamic images where first applied to medical imagery by Liang et al. [13] for breast cancer diagnosis. The authors use the dynamic image method to convert 3D digital breast tomosynthesis images into dynamic images and combined them with 2D mammography images for breast cancer classification. In our work, we propose to combine dynamic images with an attention mechanism for 3D MRI image classification.

## 3  Approach

We provide a detailed discussion of our method. First, we summarize the high-level network architecture. Second, we provide detailed information about the dynamic image method. Next, we show our classifier structure and attention mechanism. Finally, we discuss the loss function used for training.

### 3.1  Model Architecture

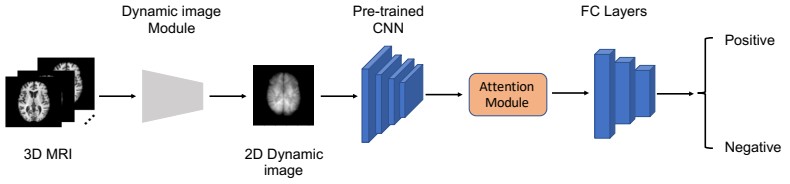

**Fig. 2.** The architecture of our 2D CNN model.

Figure 2 illustrates the architecture of our model. The 3D MRI image is passed to the dynamic image module to transform the 3D MRI image volume into a 2D dynamic image. We apply transfer learning for feature extraction with the dynamic image as the input. We leveraged a pre-trained CNN as the backbone feature extractor. The feature extraction model is pre-trained with the ImageNet dataset [5]. Because we use a lower input resolution than the resolution used for ImageNet training, we use only a portion of the pre-trained CNN. The extracted features are finally sent to a small classifier for diagnosis prediction. The attention mechanism, which is widely used in computer vision community, can boost CNN model performance, so we embed the attention module in our classifier.

## 3.2   Dynamic Image

The dynamic image method [7] [3] was originally proposed for video action recognition. For a video with T frames $I_1, ..., I_T$, the method compresses the whole video into one frame by temporal rank pooling. The compressed frame is called a dynamic image. The construction of the dynamic image is based on Fernando et al [7]. The authors use a ranking function to represent the video. $\psi(I_t) \in \Re^d$ is a feature representation of the individual frame $I_t$ of the video. $V_t = \frac{1}{t} \sum_{\tau=1}^{t} \psi(I_\tau)$ is the temporal average of the feature up to time $t$. $V_t$ is measured by a ranking score $S(t|d) = <d, V_t>$, where $d \in \Re^d$ is a learned parameter. By accumulating more frames for the average, the later times are association with larger scores, e.g $q > t \rightarrow S(q|d) > S(t|d)$ , which are constraints for the ranking problem. So the whole problem can be formulated as a convex problem using RankSVM:

$$d^* = \rho(I_1, ..., I_t; \tau) = \underset{d}{\operatorname{argmin}} E(d) \tag{1}$$

$$E(d) = \frac{\lambda}{2}||d||^2 + \frac{2}{T(T-1)} \times \sum_{q>t} \max\{0, 1 - S(q|d) + S(t|d)\} \tag{2}$$

In Equation (2), the first term is a quadratic regularization used in SVMs, the second term is a hinge-loss counting incorrect rankings for the pairs $q > t$.

The RankSVM formulation can be used for dynamic image generation, but the operations are computationally expensive. Bilen et al. [3] proposed a fast approximation for dynamic images:

$$\hat{\rho}(I_1, ..., I_t; \psi) = \sum_{t=1}^{T} \alpha_t \cdot I_t \tag{3}$$

where, $\alpha_t = 2t - T - 1$ is the coefficient associated to frame $I_t$. We take this approximate dynamic image strategy in our work for 3D MRI volume to 2D image transformation. In our implementation, the z-dimension of 3D MRI image is equal to temporal dimension of the video.

## 3.3   Classifier with Attention Mechanism

The classifier is a combination of an attention mechanism module and a basic classifier. Figure 3 depicts the structure of attention mechanism, which includes four $1 \times 1$ convolutional layers. The first three activation functions of convolutional layers are ReLU, the last convolutional layer is attached with softmax activation function. The input feature maps $I \in R^{H \times W \times C}$ are passed through the four convolutional layers to calculate attention mask $S \in R^{H \times W \times 1}$. We apply element-wise multiplication between the attention mask and input feature maps to get the final output feature map $O \in R^{H \times W \times C}$. Our basic classifier contains three fully connected (FC) layers. The output dimensions of the three FC layers are 512, 64, and 2. Dropout layers are used after the first two layers with dropout probability 0.5.

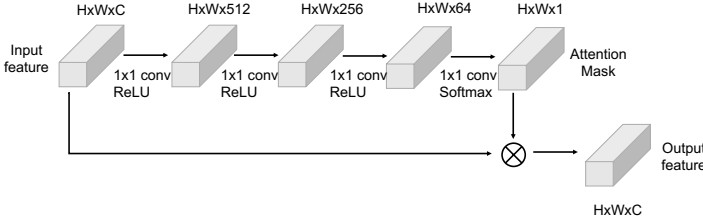

**Fig. 3.** The attention mechanism structure in our CNN model.

### 3.4 Loss Function

In previous AD classification studies, researchers mainly concentrated on binary classification. In our work, we do the same for ease of comparison. The overall loss function is binary cross-entropy. For a 3D image $I$ with label $l$ and probability prediction $p(l|I)$, the n sample loss function is:

$$loss(l, I) = -\frac{1}{n} \sum_{i=1}^{n} [l^i log(p(l|I)^i) + (1 - l^i)log(1 - p(l|I))^i] \qquad (4)$$

where the label $l = 0$ indicates a negative sample and $l = 1$ indicates a positive sample.

## 4  Evaluation

We use the publicly available dataset from the Alzheimer's Disease Neuroimaging Initiative (ADNI) [2] for our work. Specifically, we trained CNNs with the data from the "spatially normalized, masked, and N3-corrected T1 images" category. The brain MRI image size is $110 \times 110 \times 110$. Since a subject may have multiple MRI scans in the database, we use the first scan of each subject to avoid data leakage. The total number of data samples is 100, containing 51 CU samples and 49 AD samples.

The CNNs are implemented in PyTorch. We use five-fold cross validation to better evaluate model performance. The batch size used for our model is 16. The batch size of the baseline models is 8, which is the maximum batch size of the 3D CNN model trained on the single GTX-1080ti GPU. We use the Adam optimizer with $beta_1 = 0.9$ and $beta_2 = 0.999$. The learning rate is 0.0001. We train for 150 epochs. To evaluate the performance of our model, we use accuracy (Acc), the area under the curve of Receiver Operating Characteristics (ROC), F1 score (F1), Precision, Recall and Average Precision (AP) as our evaluation metrics.

### 4.1 Quantitive Results

High quality feature extraction is crucial for the final prediction. Different pre-trained CNN models can output different features in terms of size and effective

receptive field. We test different pre-trained CNNs to find out which CNN models perform best as our feature extractor. Table 1 shows various CNN models and the corresponding output feature size.

**Table 1.** The different pre-trained CNN model as feature extractors and the output feature sizes

| CNN model | Output feature size |
|---|---|
| AlexNet [12] | $256 \times 5 \times 5$ |
| VggNet11 [17] | $512 \times 6 \times 6$ |
| ResNet18 [8] | $512 \times 7 \times 7$ |
| MobileNet_v2 [15] | $1280 \times 4 \times 4$ |

Since our dynamic image resolution is $110 \times 110 \times 3$, which is much smaller than the ImageNet dataset resolution: $256 \times 256 \times 3$, we use only part of the pre-trained CNN as the feature extractor. Directly using the whole pre-trained CNN model as feature extractor will cause the output feature size to be too small, which decreases the classification performance. In the implementation, we get rid of the maxpooling layer of each pre-trained model except for the MobileNet_v2 [15], which contains no maxpooling layer. Also, because there is a domain gap between the natural image and medical image we set the pre-trained CNN models' parameters trainable, so that we can fine tune the models for better performance.

**Table 2.** The performance results of different backbone models with dynamic image as input

| Model | Acc | ROC | F1 | Precision | Recall | AP |
|---|---|---|---|---|---|---|
| AlexNet | 0.87 | 0.90 | 0.86 | 0.89 | 0.83 | 0.82 |
| ResNet18 | 0.85 | 0.84 | 0.84 | 0.86 | 0.81 | 0.79 |
| MobileNet_v2 | 0.88 | 0.89 | 0.87 | 0.89 | 0.85 | 0.83 |
| VggNet11 | 0.91 | 0.92 | 0.91 | 0.88 | 0.93 | 0.86 |

When analyzing MRI images using computer-aided detectors (CADs), it is common to strip out the skulls from the brain images. Thus, we first test the proposed method using the MRI with the skull stripped. Our proposed model takes dynamic images (Dyn) as input, VGG11 as feature extractor, and a classifier with the attention mechanism: $Dyn + VGG11 + Att$. The whole experiment can be divided into three sections: the backbone and attention section, the baseline model section, and the pooling section. In the backbone and attention section, we use 4 different pre-trained models and test the selected backbone with and with-

**Table 3.** The performance results of different 2D and 3D CNN models

| Model | Acc | ROC | F1 | Precision | Recall | AP |
|---|---|---|---|---|---|---|
| 3D-VGG [11] | 0.80 | 0.78 | 0.78 | 0.82 | 0.75 | 0.74 |
| 3D-ResNet [11] | 0.84 | 0.82 | 0.82 | 0.86 | 0.79 | 0.78 |
| Max. + VGG11 | 0.80 | 0.77 | 0.80 | 0.78 | 0.81 | 0.73 |
| Avg. + VGG11 | 0.86 | 0.84 | 0.86 | 0.83 | 0.89 | 0.79 |
| Max. + VGG11 + Att | 0.82 | 0.76 | 0.82 | 0.80 | 0.83 | 0.75 |
| Avg. + VGG11 + Att | 0.88 | 0.89 | 0.88 | 0.85 | **0.91** | 0.82 |
| Ours | **0.92** | **0.95** | **0.91** | **0.97** | 0.85 | **0.90** |

out the attention mechanism. Based on the performance shown in Table 2, we choice VGG11 as the backbone model. In the baseline model section, we compare our method with two baselines, namely 3D-VGG and 3D-ResNet. Table 3 shows the performance under different CNN models. The proposed model achieves 9.52% improvement in accuracy and 15.20% better ROC over the 3D-ResNet. In the pooling section: we construct two baselines by replacing the dynamic image module with the average pooling (Avg.) layer or max pooling (Max.) layer. The pooling layer processes the input 3D image over the z-dimension and outputs the same size as the dynamic image. Comparing with the different 3D-to-2D conversion methods under the same configuration, the dynamic image outperforms the two pooling methods.

## 4.2 Pre-processing Importance Evaluation

**Table 4.** The performance results of different 2D and 3D CNN models on the MRI image with skull.

| Model | Acc | ROC | F1 | Precision | Recall | AP |
|---|---|---|---|---|---|---|
| 3D-VGG [11] | 0.78 | 0.62 | 0.77 | 0.80 | 0.75 | 0.72 |
| Ours | 0.63 | 0.52 | 0.63 | 0.62 | 0.64 | 0.57 |

In this section, we show results using the raw MRI image ( including skull ) as input. We perform experiments on the same patients' raw brain MRI image with the skull included to test the performance of our model. The raw MRI image category is "MT1,GradWarp,N3m". The image size of the raw MRI image is "$176 \times 256 \times 256$". Figure 4 illustrates the dynamic images of different participants' MRI brain images with the skull. The dynamic images are blurrier than the images under skull striping processing. This is because the skull variance can be treated as noise in the dynamic image. Table 4 shows the significant performance decrease when using 3D Brain MRI images with skull. Figure 4 shows

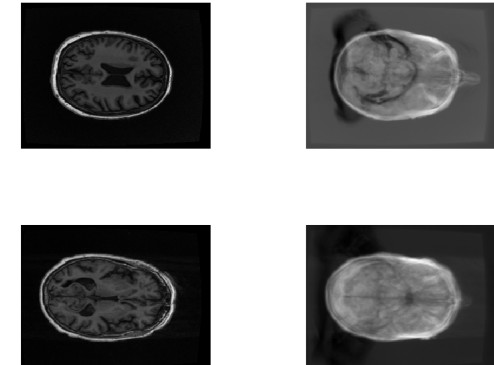

**Fig. 4.** The original MRI image with skull and its correspond dynamic image. The first row shows the MRI sample slice of a CU participant and the dynamic image over the z-dimension pooling. The second row shows the MRI sample slice of an AD participant and its corresponding dynamic image.

a visual representation of how the dynamic images are affected by including the skull in the image. In this scenario, the model can not sufficiently diagnose the different groups. A potential cause of this decrease if performance is that the dynamic image module is a pre-processing step, and the module is not trainable. We believe an end-to-end, learnable approximation for dynamic images would improve performance.

### 4.3   Models Training time

**Table 5.** The total 150 epochs training time of different CNN models.

|  | Training time(s) |
|---|---|
| 3D-VGG [11] | 2359 |
| 3D-ResNet [11] | 3916 |
| Ours | 414 |

Another advantage of the proposed model is faster training. We train all of our CNN models for 150 epochs on the same input dataset. Table 5 shows the total training time of the different 2D and 3D CNN models. Compared with the 3D-CNN networks, the proposed model trains in about 20% of the time. Also, due to the higher dimension of the 3D convolutional layer, the number of parameters of the 3D convolutional layer is naturally higher than the 2D convolutional layer. By applying the MobileNet [9] or ShuffleNet [19] in medical

image diagnosis, there is potential for mobile applications. We used MobileNet for our experiments. We used the MobileNet v1 achitecture as the feature extractor and obtained 84.84% accuracy, which is similar in accuracy to the 3D ResNet.

## 5    Conclusions

We proposed to apply the dynamic image method to convert 3D Brain MRI images into 2D dynamic images as the inputs for a pre-trained 2D CNN. The proposed model outperforms a 3D CNN with much less training time and improves 9.5% better performance than the baselines. We trained and evaluated on MRI brain imagery and found out that brain skull striping pre-processing is useful before applying the dynamic image conversion. We used an offline dynamic image module in our experiments, but we believe it would be interesting to explore a learnable dynamic image approximation in the future. We showed that combining the dynamic image with pre-trained efficient networks, and found that they performed similarly to the 3D CNN model.

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
