# OpenReview forum: "Dynamic Image for 3D MRI image Alzheimer’s Disease classification"
_thecvf.com/ECCV/2020/Workshop/BIC — BIC 2020 Oral_

### Official Review · AnonReviewer3 · 2020-07-30
**An interesting moslty solid contribution with some problematic unclear aspects.**

**Rating:** 5
**Confidence:** 4

**Review:**

-----------------------------
Summary
-----------------------------
The authors present a novel method for the diagnosis of Alzheimer's disease from MRI volumes.
The first compress the 3D volume to a 2D 'dynamic' image, which is then fed into a pre-trained features extractor.
Finally, the resulting feature vector is subject to an attention mechanism and processed by a fully connected MLP to predict a probability for the disease.
The method is evaluated against various baselines and achieves competitive results.

-----------------------------
Strengths
-----------------------------

- The paper is for the most part clearly written and structured.
- The method outperforms its baselines, which are adequatly choosen as far as I can tell.

-----------------------------
Weaknesses
-----------------------------

My main concern is the unclear explanation of the dynamic image compression.
This would not be a big issue, since they say they are using an existing method [3].
However, looking at [3], I am not sure the equation they give (Eq. 3) corresponds to what is presented in [3] and if it is really correct.
To be more precise, Eq 3 computes simply an averaged image using fixed weights \alpha_t. It does not even make use of the feature representation.
In contrast, the method in [3] averages the computed feature representations, as far as I understand.
I am really not sure if this is just a typo in Eq 3 or if the authors are simply averaging the images with fixed weights and overselling it as a more sophisticated method.

The mathematical notation is not very clear.
In line 140 'd' is used as the dimension of a feature vector. In line 143 it is itself a feature vector.
In Section 3.2 the 'I' denotes an input image, but in Section 3.3  the same symbol is used for the feature image produced by the network.
Throughout the paper, various symbols are used for the set of real numbers.

-----------------------------
Final Recommendation
-----------------------------

Even though the paper seems highly relevant and generally solid, I am really concerned about the computation of the dynamic image.
I thus see the paper as borderline.
However, as I am not an expert on dynamic images, it might be that I simply misunderstood this part or that it can be explained as a typo.


**Reviews Visibility:**

I agree that my anonymized review is made publicly visible, if the submission is accepted.

---

### Official Review · AnonReviewer1 · 2020-07-31
**Simple and effective method to improve AD classification**

**Rating:** 8
**Confidence:** 4

**Review:**

Summary
-------

This paper presents a method for Alzheimer's Disease (AD) classification from
3D MRI scans. In contrast to earlier approaches, the 3D scans are first
converted into a 2D dynamic image, which is then processed by a 2D
convolutional neural network. The network consists of a pre-trained (and
fine-tuned) feature extractor, an attention module, and subsequent fully
connected layers for the final classification. Results on MRI scans with
stripped skulls demonstrate a significant increase compared to fully 3D
baselines, albeit using only 20% of the time needed for training.

Quality and Clarity
-------------------

The introduction, motivation, and technical description are very well written
and easy to follow. The provided figures are helpful to understand the method
and show qualitative results of the dynamic image generation.

Originality
-----------

Using 2D dynamic images instead of the full 3D scan is a compelling idea
(although already explored in other medical domains). The main contribution of
this paper lies therefore in the subsequent use of pre-trained 2D feature
extractors, which are enabled by the 2D input images. Furthermore, the authors
investigate the usefulness of an attention module between the feature
extraction and classification network.

Significance
------------

The results demonstrate consistent improvements over several baselines and
variations of the proposed model. On top of those improvements, the proposed
model trains faster and requires less resources for prediction.

The authors mention that "there is potential for mobile applications" (line
360). I am not sure what this could look like in a clinical setting and suggest
the authors elaborate if they want to make this point convincingly.

My main concern for a clinical application is that the improved results seem to
be achieved only on "skull stripped" MRI scans. From the paper, it is unclear
whether this is a manual, semi-automatic, or entirely automatic process. I
would appreciate if the authors could discuss this point in more detail to
provide context about the amount of manual labor needed in the proposed
pipeline.

Pros
----

* well written
* elegant architecture
* thorough comparison to baselines and model variations
* significant accuracy improvement

Cons
----

* unclear how much manual intervention is needed for "skull stripping"

Minor Comments
--------------

* title: capitalize "image" and "classification"
* line 56: "We" -> "we"
* line 144: "association" -> "associated"
* line 283: "choice" -> "choose"
* line 335: "decrease if" -> "decrease in"
* Equation 4: What is $i$ running over? According to the text, $l$ and $I$ are label and image of a single sample.
* Figure 4: It would be helpful to see same (or similar) images without skull in the same figure.
* notation: $\mathcal{R}$ vs $R$


**Reviews Visibility:**

I agree that my anonymized review is made publicly visible, if the submission is accepted.

---

### Official Review · AnonReviewer2 · 2020-07-31
**Dynamic Image for 3D MRI image Alzheimer's Disease classification**

**Rating:** 8
**Confidence:** 4

**Review:**

Summary
This paper addresses the problem of 3D MRI volume segmentation of images collected from patients with Alzheimer's disease. The motivation for the work comes from automating the classification task for assigning labels such as cognitive unimpaired (CU) and Alzheimer’s disease (AD) to each patient based on the 3D MRI volume. The authors approach the problem by  leveraging a 3D to 2D conversion using dynamic images according to https://openaccess.thecvf.com/content_cvpr_2015/papers/Fernando_Modeling_Video_Evolution_2015_CVPR_paper.pdf and https://www.egavves.com/data/cvpr2016bilen.pdf and introducing an attention module in a transfer model with pre-training on ImageNet dataset. The evaluations include (a) matching feature dimensionality of the features extracted from four well-established architectures and the features coming from the dynamic image based conversion, (b) optimizing the steps in feature extraction and AI-based classification, (c) analyzing inclusion/exclusion of skull, and (d) comparing execution times when 3D vs 2D raw data are used as inputs into classifiers.
Strengths:
The classification framework is very interesting.
The introduction of dynamic image and attention module is novel

Weaknesses:
The experimental dataset is very limited.
The theoretical description is not very clear.

Comments:
Is there any reason (e.g., based on visual inspection) to believe that the features of 2D dynamic images are of the same nature as the features extracted from ImageNet?
Line 129: what do you refer to when mentioning the ImageNet resolution?
Line 171: why did you choose three activation functions and 1x1 convolutional kernels?
The section 3.3 also did not explain the details in Fig 3. For example, why do you have in Fig 3 the same blocks but the tensor sizes are HxWx512 -> HxWx256 -> HxWx64? Shouldn’t the last tensor size be HxWx128?
What implementation did you use for the CAM attention module?
What is the method in dynamic image based conversion that you are using to create 110 x 110 x 3 (i.e., 3 features)? The original paper in https://openaccess.thecvf.com/content_cvpr_2015/papers/Fernando_Modeling_Video_Evolution_2015_CVPR_paper.pdf refers to learning rank machines but your paper does not mention this important detail.


Minor comments:
Fig 1 caption: you have one row of pictures but the caption refers to two rows.
Why applying dynamic image-based 3D to 2D conversion is preferred over z-axis? Is there any motivation to prefer one of the three possible planes, i.e., sagittal vs transversal vs coronal plane?



**Reviews Visibility:**

I agree that my anonymized review is made publicly visible, if the submission is accepted.

---

### Decision · Program_Chairs · 2020-07-31

Accept (Oral)